# Toward Digital Twin Development for Implant Placement Planning Using a Parametric Reduced-Order Model

**DOI:** 10.3390/bioengineering11010084

**Published:** 2024-01-16

**Authors:** Seokho Ahn, Jaesung Kim, Seokheum Baek, Cheolyong Kim, Hyunsoo Jang, Seojin Lee

**Affiliations:** 1Department of Digital Manufacturing, Hanbat National University, 125 Dongseo-daero, Yuseong-gu, Daejeon 34158, Republic of Korea; ash93@hanbat.ac.kr (S.A.); leesj818@hanbat.ac.kr (S.L.); 2Department of Industry-Academic Convergence, Hanbat National University, 125 Dongseo-daero, Yuseong-gu, Daejeon 34158, Republic of Korea; 3Digital Platform Team, DNDE Inc., Busan 48059, Republic of Korea; shbaek@dnde.co.kr; 4Implant Research Laboratory, Cybermed 6-26, Yuseong-daro 1205 beon-gil, Yuseong-gu, Daejeon 34104, Republic of Korea; kaster@cybermed.co.kr (C.K.); jhsoooo@cybermed.co.kr (H.J.)

**Keywords:** finite element analysis, computer-aided engineering, implant, reduced-order model, 1-D CAE solver, digital twin

## Abstract

Real-time stress distribution data for implants and cortical bones can aid in determining appropriate implant placement plans and improving the post-placement success rate. This study aims to achieve these goals via a parametric reduced-order model (ROM) method based on stress distribution data obtained using finite element analysis. For the first time, the finite element analysis cases for six design variables related to implant placement were determined simultaneously via the design of experiments and a sensitivity analysis. The differences between the minimum and maximum stresses obtained for the six design variables confirm that the order of their influence is: Young’s modulus of the cancellous bone > implant thickness > front–rear angle > left–right angle > implant length. Subsequently, a one-dimensional (1-D) CAE solver was created using the ROM with the highest coefficient of determination and prognosis accuracy. The proposed 1-D CAE solver was loaded into the Ondemand3D program and used to implement a digital twin that can aid with dentists’ decision making by combining various tooth image data to evaluate and visualize the adequacy of the placement plan in real time. Because the proposed ROM method does not rely entirely on the doctor’s judgment, it ensures objectivity.

## 1. Introduction

A digital twin connects physical and digital objects in real time and utilizes simulation methods to support decision making for optimizing the functions of physical objects. Currently, digital twins are used in the manufacturing, automobile, and energy industries and have been developed into a form that integrates physical and virtual spaces via continuous information transmission [1,2,3]. Recently, digital twins were adopted in the medical field to build an innovative process for simultaneously performing disease diagnosis, prognosis, and treatment [4,5]. In the field of dentistry, the digital twin technology is gradually being applied to advance digital dentistry [6]. Therefore, a method that can improve the prognosis after treatment by predicting results via real-time simulations using patient data is highly desired.

Dental implants are used in dental treatment to replace the teeth of partially and completely edentulous patients [7]. The implant placement process involves placing an implant in the jawbone, combining the implant with a small metal pillar, and inserting a tooth-shaped prosthesis into the oral cavity [8,9]. The 10-year survival rate of implants is more than 90%, and with proper management, they can be used for decades or more [10]. However, resorption of the bone and other medical complications have been observed in some cases. These issues can be attributed to factors such as trauma during implant surgery, bacterial infiltration in the micro gap, and tissue deformation around the implant [11,12,13]. Many clinical trials are currently underway to solve this problem [14,15]. Further, implant stability has been determined by confirming the degree of the osseointegration of the implant via resonance frequency analysis [16,17,18].

Existing methods are employed after implant placement; therefore, the abovementioned problems cannot be identified in advance, which makes determining the stress distribution difficult. Finite element analysis (FEA), which can be used to obtain approximate solutions via partial differential equations using the finite element method (FEM), has been used to solve complex numerical problems [19]. FEA can also determine the stress distribution and deformation that occurs in implants and bones [20]. Such stress distribution and deformation data are utilized to verify stability via a stress structure analysis before implant placement [21,22].

Factors such as the diameter and length of the implant, placement location, left–right angles, front–rear angles, Young’s modulus of the cancellous bone, and thickness of the cortical bone can affect implants [23,24,25,26,27]. The effect and stability of each of these factors should be determined because they can introduce variations in the results such as osseointegration. The largest force is applied to the cortical bone, which has a high Young’s modulus, because the force acts on the top of the implant owing to masticatory and dental grip forces [28,29]. Further, stability is evaluated based on the Von mises stress that occurs in the cortical bone because this area has the most nerves [30].

Previous studies have reported a decrease in stress values in the cortical bone with an increase in the implant length and thickness [31,32,33]. Further, the generated stress was reported to increase with an increase in the front–rear and left–right angles of the implant [34,35,36]. The implant placement location has been divided into anterior, premolar, and posterior teeth, and the magnitude of stress has been reported to decrease in the order of posterior teeth, premolars, and anterior teeth [37]. Although many studies have focused on implant stress for each variable, to the best of our knowledge, no study analyzes all variables simultaneously.

This study aims to improve the post-placement success rates of dental implants by making real-time stress distribution data for implants and cortical bones available. To this end, for the first time, the six important variables related to establishing an implant plan were analyzed simultaneously to derive results for all possible cases. A real-time simulation was performed using a parametric reduced-order model(ROM) instead of deriving the results obtained by analyzing all cases before implantation. Parametric ROM is a mathematical modeling technique that can solve the problem of inefficiency in the analysis time attributed to the complexity and high degree of freedom of a model. Herein, 46,787 cases were created by combining the six variables, and they were simplified to 25 cases via the design of experiment (DOE). Subsequently, the results were obtained via analysis. These results were used to obtain a 1-D CAE solver from the ROM by considering factors that afforded the maximum accuracy via the coefficient of determination (CoD) and coefficient of prognosis (CoP). Thus, the stress and deformation values in the cortical bone and implant could be calculated instantaneously to evaluate stability.

The developed model was installed in ONDemend3D: version 1.0.11.1007 commercial software that collects and processes existing dental image data, identifies areas of missing teeth, inspects labeling, and projects cone-beam computed tomography images in two dimensions. The program automatically calculates and visualizes the stress value of the implant and cortical bone using patient information such as bone tissue and angle of implant placement.

In contrast to existing implant placement plans determined based on the doctor’s discretion and data obtained via X-ray, facial, and intra-oral photographs, an optimal location and installation angle for implants can be suggested using objective indicators with the proposed approach. Moreover, a function is developed to analyze and visualize the stress and deformation information in real time based on the setting and changes in the long axis of the implant. The real-time stress value calculation and visualization functions are expected to play important roles in the development of digital twins that can aid dentists in determining appropriate implant placement plans.

## 2. FE Modeling Parameters and Boundary Conditions

In the implantation procedure, a hole is first drilled into the bone to place an implant. The bone and implant combine after three to six months, and a crown is finally placed on top of the bone. The implant can be used for at least 10 years after placement; therefore, it should cause minimal stress on the cortical bone, where many nerves are also present. Thus, the safest approach is to plant the implant vertically at a position where the cortical bone is the thickest. However, in a few instances, the implant placement must be optimized for the shape of the surrounding teeth and bones.

### 2.1. Implant Model

#### 2.1.1. Basic Model Description

Figure 1 shows a 3D computer-aided-design (CAD) model of the implant and bone. The shape of the bone was simplified to the most common CAD model and divided into cortical and cancellous bones. An implant commonly uses a macro thread, which is a specification that has been adopted by many companies. Therefore, a hole was drilled to reproduce the state after the actual implant procedure once the implant was inserted into fully combined cortical and cancellous bones.

#### 2.1.2. Boundary Conditions

The contacts used for the three bodies are presented in Figure 2a. Cortical and cancellous bones are single bones, and therefore, a bonded contact is used in which the x, y, and z directions are constrained; the contact length/area does not change, and separation is not allowed. In addition, a frictional contact that transmits shear stress at a specific ratio is used for the implant and cortical bone and for the implant and cancellous bone because the screw is attached to the contact surface. The friction coefficient between the bone and implant is set to 0.2.

The model shown in Figure 2b is analyzed considering the load conditions of the dental grip force. Dental force is generated when the upper and lower teeth come in contact and act in the vertical direction. A force of 100 N was applied downward from the top of the abutment. When the implant is placed at an angle owing to interference from neighboring teeth, the crown surrounding the abutment applies a force in the vertical direction on the lower end in global coordinates rather than on the lower end in the local direction of the implant.

Finally, the CAD model was expressed by cutting the bone to a size that could accommodate an implant. Therefore, the degrees of freedom in the x, y, and z directions were constrained by a fixed support placed on the bone in the direction of the tooth.

#### 2.1.3. Material Properties

Table 1 lists the physical properties considered in this analysis. The implant used in this study is composed of titanium, and therefore, it has a large modulus of elasticity and Poisson’s ratio. The modulus of elasticity is 13 GPa for the cortical bone, which is hard and has many nerves. The analysis was conducted using a basic elastic modulus of 7.0 GPa for the cancellous bone, which varies based on the patient [38].

### 2.2. Implant Region and Design Variables

#### 2.2.1. Regions of Anterior, Premolar, and Posterior Teeth

Teeth, including the thin and small front teeth to the thick and large inward teeth, are referred to by different names, as shown in Figure 3 [39]. In this study, bones are classified as anterior teeth, premolars, and posterior teeth, as shown in Figure 4.

Anterior teeth are the small and thin teeth from the central incisors to the cuspids that are used to cut and tear food. Premolars are the two teeth that exist between the anterior and posterior teeth. They are used to break down food particles, and they have a shape similar to that of the anterior teeth; however, they are thicker. Finally, posterior teeth are areas where all the molars are combined. These teeth have a flat shape, and they are only used for crushing and grinding. These teeth have a shape different from that of the other bones and are thicker. The thicknesses of the cortical bone of the anterior, premolar, and posterior teeth used for the analysis were 1.5, 2.0, and 2.5 mm, respectively.

#### 2.2.2. Implant Length

Figure 5 indicates that an analysis is performed for five implant lengths: 8.5, 10, 11.5, 13, and 15 mm. Implants with lengths in the range of 8.5–11.5 mm are used for the posterior teeth, and those in the range of 10–15 mm are used for the anterior teeth. Although shorter implants exist, they are not used because of safety concerns. The lower part of a neural tube must be considered in the lower jaw. In the upper jaw, there are cases where the gum bone is insufficient, and therefore, implants exceeding 15 mm are not used often.

#### 2.2.3. Implant Diameter

Figure 6 shows that an analysis is performed using five implant diameters of 3.0, 3.5, 4.0, 4.5, and 5.0 mm. Implants with diameters in the range of 3.0–3.5 mm are used in the anterior teeth, and implants with diameters of 4.0 mm or more are used in the posterior teeth. The larger the diameter, the safer is the result. Implants with different diameters are used based on the scenario because an incorrect implant can invade the major anatomical structures around it and cause damage.

#### 2.2.4. Young’s Modulus of the Cancellous Bone

Table 2 lists the Young’s modulus of the cancellous bone. In contrast to the hard cortical bone, the cancellous bone has several different physical properties. For example, it has a modulus of elasticity of 5.5–9.5 GPa in the anterior region and 0.5–2.0 GPa in the posterior region; these values vary based on the individual in question. At a young age, the bone density is high, and therefore, it exhibits a high elastic modulus. However, the elastic modulus of the cancellous bone tends to decrease with age-related deterioration (decrease in bone density). We investigated the stability in various cancellous bones using Young’s moduli in the range of 0.5–9.5 GPa.

#### 2.2.5. Front–Rear Angle

Figure 7 shows that an analysis is performed by changing the implant angles to 0, 2.5, 5, 7.5, and 10°. The front–rear angle indicates the rotation inside or outside the oral cavity based on the center of the implant and teeth. The implant plan may use a curved abutment that can stand vertically when the implant cannot be placed vertically because of the surrounding structures; the implant can therefore be tilted at angles exceeding 10°. In this case, the tendency of the stress acting on the cortical bone is different. In this study, the analysis was conducted only for angles up to 10°, which is the maximum range that does not require the use of a curved abutment.

#### 2.2.6. Left–Right Angle

Left–right angles were analyzed by changing the implant angles to 0, 2.5, 5, 7.5, and 10°, as shown in Figure 8. The left–right angle is rotation in the direction of the oral tooth based on the center of the implant and teeth. In the left and right directions, the stress tendency is evaluated up to an angle of 10° and thus does not use an inclined abutment.

#### 2.2.7. Cortical Bone Thickness

As shown in Figure 9, an analysis is performed by changing the thickness of the cortical bone to 1.5, 1.75, 2.0, 2.25, and 2.5 mm. The cortical bone covers the soft cancellous bone. The size of the cortical bone varies for each individual, and the bone may be lost during tooth extraction. Therefore, the stress tendency is determined by varying the thickness of the cortical bone.

## 3. Numerical Results (Finite Element Analysis Results)

The von Mises stress in the cortical bone is a factor that directly affects humans. Figure 10 shows the von Mises stress results for the implant and cortical bone. The analysis was conducted using the basic model without changing the angle or thickness. The largest von Mises stress occurred at the top where the screw threads of the implant and cortical bone contacted each other; the von Mises stress decreased toward the bottom of the screw.

Therefore, the tendency of the output was evaluated via a preliminary survey before generating the DOE for each variable. The output was not linear because the input variables were not completely linear, and the input variables remained in non-linear contact with the screw. However, the accuracy of the ROM results inevitably decreases if wave-type results are obtained without a clear effect on numerical change. In the analysis, only the tendency of the cortical bone was considered because implants made of titanium are not as severely affected by stress as the cortical bone.

### 3.1. Structural Analysis Results Based on Implant Length

An analysis was conducted based on the implant length. The radius, Young’s modulus of the cancellous bone, left–right angle, front–rear angle, and cortical bone thickness were set as 4.0 mm, 7.0 GPa, 0°, 0°, and 1.5 mm, respectively, and these values were used as the criteria to ensure that the same conditions were applied. The analysis was conducted for a minimum thickness range of 8.5–15 mm. The largest stress occurred at the first point where the implant screw and bone met. The results are summarized in Figure 11 and Table 3. The magnitude of the implant stress acting on the bone decreases with an increase in the length of the implant. This is because a longer implant allows the stress to be shared across a wider area of the bone.

### 3.2. Structural Analysis Results Based on Implant Diameter

An analysis was performed by changing the thickness from 3.0 to 5.0 mm. The implant length, Young’s modulus of the cancellous bone, left–right angle, front–rear angle, and thickness of the cortical bone were set as 11.5 mm, 7.0 GPa, 0°, 0°, and 1.5 mm, respectively, to ensure that the same conditions were used. In this case, the largest stress occurred at the first point where the implant screw and bone met; the results are summarized in Figure 12 and Table 4. The surface area increases with an increase in the thickness of the implant, which results in decreased stress.

### 3.3. Structural Analysis Results Based on the Young’s Modulus of the Cancellous Bone

An analysis was performed by changing the Young’s modulus of the cancellous bone. The implant length, radius, left–right angle, front–rear angle, and thickness of the cortical bone were set as 11.5 mm, 4.0 mm, 0°, 0°, and 1.5 mm, respectively, to ensure similar conditions. Figure 13 and Table 5 indicate that the stress received by the cancellous bone increases and the stress in the cortical bone decreases with an increase in the elastic modulus of the cancellous bone.

### 3.4. Structural Analysis Results Based on the Front–Rear Angle

An analysis was performed by changing the angle from 0° to 10° in the inward and outward directions of the oral cavity. The implant length, radius, Young’s modulus of the cancellous bone, left–right angle, and thickness of the cortical bone were set as 11.5 mm, 4.0 mm, 7.0 Gpa, 0°, and 1.5 mm, respectively, to ensure similar conditions. In this case, the largest stress occurred in the curved part because the force was received in the oblique direction of the bone and not in the area where the implant screw thread and bone first met. Figure 14 and Table 6 indicate that the stress increased with an increase in the implant placement angle.

### 3.5. Structural Analysis Results Based on the Left–Right Angle

An analysis was performed by changing the direction of the teeth from 0° to 10°. The implant length, radius, Young’s modulus of the cancellous bone, front–rear angle, and thicknesses of the cancellous, front–rear, and cortical bones were set as 11.5 mm, 4.0 mm, 7.0 GPa, 0°, and 1.5 mm, respectively, to ensure that the conditions remained the same. The highest stress occurred in the bending direction with a change in the angle. As shown in Figure 15 and Table 7, the stress acting on the cortical bone increased with an increase in the angle, similar to that for the front–rear angle.

### 3.6. Structural Analysis Results Based on Cortical Bone Thickness

An analysis was performed by changing the thickness of the cortical bone from 1.5 to 2.5 mm. The implant length, radius, Young’s modulus of the cancellous bone, left–right angle, and front–rear angle were set to 11.5 mm, 4.0 mm, 7.0 GPa, 0°, and 0°, respectively, to ensure that the conditions were the same. Further, there was no clear trend according to the changes in the cortical bone thickness. The stress on the cortical bone was distributed with an increase in the thickness; however, the size of the cancellous bone decreased, resulting in a decrease in the Young’s modulus. Therefore, the results canceled each other. Figure 16 and Table 8 show that the results change with the thickness. The thickness variable of the cortical bone affected the accuracy of the results during ROM production because the deviation was severe.

### 3.7. Selecting Important Design Variables

The analysis criteria were set to the most commonly used criteria in actual implants: length = 11.5 mm; radius = 4.0 mm; Young’s modulus for the cancellous bone = 7.0 GPa; left–right and front–rear angles = 0°; and cortical bone thickness = 1.5 mm.

The differences between the largest and smallest stress values for the implant length, thickness, Young’s modulus of the cancellous bone, front–rear angle, and left–right angle were 30.0%, 57.9%, 71.9%, 42.5%, and 42.0%, respectively. The cortical bone thickness did not show a clear trend. Although there may be some errors because of differences in the scales of the input variables in each condition, the results of selecting the minimum and maximum values used in the implant process significantly affect the Young’s modulus of the cancellous bone. The order of influence on the stress occurring in the cortical bone is as follows: Young’s modulus > implant thickness > front–rear angle > left–right angles > implant length.

## 4. Simulation Data and Parametric ROM for Implant Placement Planning

The proposed ROM model was employed to connect implant simulation data to images and oral scan data for implant placement guidance. The ROM reproduces a high-precision analysis model and enables 3D simulations for use as simple mathematical models or embedded software. General ROM theories include data-based metamodel methods and physics-based proper orthogonal decomposition (POD), singular value decomposition (SVD), and Karhunen–Loeve decomposition methods.

In this study, six design variables, including implant length, radius, front–rear angle, left–right angle, Young’s modulus of the cancellous bone, and cortical bone thickness for the anterior, premolar, and posterior teeth were used to enable real-time 3D implant placement planning.

### 4.1. Simulation Data Generation Using Design of Experiment

The Taguchi orthogonal array, volume 5, level L25, was used for the study. Figure 17 shows the six design variables and the ranges required for implant placement planning. Figure 18 shows an example of the experimental arrangement (design matrix) and FEA results of the L25 ($5^{6}$) orthogonal array table for the six design variables.

### 4.2. Parametric ROM

Model reduction and surrogate modeling are used to obtain outputs in real time for large complex systems or specific inputs in a quick and efficient manner. A parametric ROM is used to create a reduced model to respond to the changes in parameters constituting a system. Therefore, the parametric ROM involves fitting a reduced model or reducing the bias based on input parameters such as shape, load, and boundary conditions.

In this study, the parametric ROM employed for implant placement planning used high-fidelity polynomials, support vector regression (SVR), genetic algorithm for the identification of a robust subset (GARS), and deep feature fusion network (DFFN). The DFFN implements a metamodel based on the artificial neural network (ANN) paradigm using the Keras library with TensorFlow as the back-end to create a metamodel by building and training a network [40,41,42].

The quality of the parametric ROM was evaluated using the CoD and CoP.
(1)$$CoD=\frac{\sum_{k=1}^{N}{\left({\hat{y}}_{j}^{\left(k\right)}\left(x_{1},x_{2},\cdots ,x_{j},\cdots ,x_{n}\right)-\mu_{y_{j}}\right)}^{2}}{\sum_{k=1}^{N}{\left(y_{j}^{\left(k\right)}\left(x_{1},x_{2},\cdots ,x_{j},\cdots ,x_{n}\right)-\mu_{y_{j}}\right)}^{2}}\;$$

The CoD can be used to assess the approximation quality of a polynomial regression model. This measure is defined as the relative amount of variation explained by the approximation [43,44]
(2)$$CoP=[\frac{E{\left[Y_{Test}{\hat{Y}}_{Test}\right]}^{2}}{\sigma Y_{Test}\sigma {\hat{Y}}_{Test}}{]}^{2}=[\frac{\sum_{k=1}^{N}\left(y^{\left(k\right)}-\mu_{y}\right)\left({\hat{y}}^{\left(k\right)}-\mu_{y}\right)}{\left(N-1\right)\sigma_{y}\sigma_{\hat{y}}}],0\leq CoP\leq 1$$
where y represents the test value, $\hat{y}$ represents the estimate of the parametric ROM, and σ and μ represent the mean and standard deviation, respectively, calculated from each of the N-many DOE data. The degree of agreement between the DOE data and the estimate of the parametric ROM was measured using the CoP value in Equation (Equation 2) for additional test design points [45,46].

The CoP $y^{\left(k\right)}$ is calculated in a similar manner as the more common CoD or $R^{2}$ values; however, it is calculated via a cross-validation process whereby the data are partitioned into subsets used either for the metamodel or CoP calculation. Therefore, the CoP is preferred as a measure of the effectiveness of a model for predicting unknown data points, which is valuable for this type of parametric ROM application.

The generalized CoD can be applied to all parametric ROMs and is equivalent to the square of the linear correlation coefficient between the true sample values and model predictions. The quality of the prognosis must be evaluated to determine the quality of an approximation; therefore, an additional test dataset is used, and the agreement between these real test data and parametric ROM estimates are measured using the CoP.

Figure 19 shows the approximation accuracy of the high-fidelity surrogate model. GARS is a parametric ROM with high approximation quality. The CoD of the three implant areas is greater than 97% for both stress and strain; the stress values of the most important cortical bone for arbitrary design points are approximately 90%, 88%, and 97% for the anterior, premolar, and posterior regions, respectively, which indicates good quality.

Figure 20 shows the polynomial equations used to calculate the stress and strain values for the anterior, premolar, and posterior regions for real-time implant placement planning that requires selecting the order of importance of design variables and considering interactions. The approximation quality in all areas is approximately 96% or higher.

Figure 21, Figure 22 and Figure 23 compare the implant FEA results and data-driven parametric ROM based on the anterior, premolar, and posterior regions, respectively. Although there was an error in the approximation, the location and analysis of the ROM prediction values were approximated effectively.

Table 9 presents a comparison of the time required to obtain the results via FEA and parametric ROM. In contrast to FEA, which requires 2 h and 25 min to obtain the results, parametric ROM can obtain the analysis results from the 1-D CAE solver in 1 s by entering the variables.

## 5. OnDemand3D Program for Real-Time Implant Placement Guide

The landscape of dental implant technology is continually evolving. The current trend involves integrating intra-oral scan data and digital twin technologies into implant planning software.

This technology enables meticulous pre-surgical planning (planning the precise location, angle, and depth of implant placement) using 3D imaging and digital twin models. During the implantation procedure, a skilled dentist needs a plan to establish the location and angle of implantation. Parametric ROM technology provides real-time, physical behavior (bone strength around the implant), and visual guidance to doctors using CT data to locate a suitable implant position and select the necessary implants.

The 1-D formula obtained using structural analysis and ROM data was applied to the OnDemand3D software, which is highly advanced 3D imaging software developed for dentists, clinicians, and research experts for use in planning and simulation of patient treatment, accurate diagnosis, and advanced research. OnDemand3D utilizes DICOM data across modalities to reconstruct 3D volumes using the latest and best in 3D technology. OnDemand3D provides specialized layouts, reconstructed images, and tools for accurate and precise diagnoses.

The software consists of modules such as demineralized bone matrix, Dental, 3D, 3D Cephalometric, Fusion, Dynamic LightBox, Report, Xlmage, and In2Guide, and each module functions independently. The dental module applied in this study is the standard for 3D dental reconstruction. This module contains eight different layouts, including dental, verification, multiplanar reformation (MPR), panorama, temporomandibular joint (TMJ), bilateral-TMJ, orthodontic, and airway. As shown in Figure 24, implant surgery can now be fully simulated using the dental module with the help of our real-size library of implant fixtures and abutments as well as the intra-oral/3D model scan data alignment function.

This module plays an auxiliary role for the user by integrating the implant ROM model to provide additional information. All necessary parameters from the implant ROM can be obtained from this module. As shown in Figure 25, the implant to be used is selected from the implant library by selecting the location of the tooth to be installed. Based on the selected location, appropriate bone material property values can be obtained, as shown in Figure 26.

As shown in Figure 27, the empty space between the teeth is determined from the photographic data, and the implant is placed. The six variables are subsequently changed, as shown in Figure 28, and are input in the 1-D CAE solver for automatic calculation. The stress level in the cortical bone is displayed in green for under 40 MPa and in red for higher.

In the future, we plan to subdivide the range based on the experiences of actual dentists and present it as a gradation from safe (green) to dangerous (red).

## 6. Summary Remarks and Conclusions

Stress overload that occurs during implant placement can lead to osseointegration and failure [47]. Therefore, many studies used FEA to reduce overload before implant placement. Most studies conduct stress analysis based on new implants [48,49], changes in implant materials [50,51,52], and various variables [53]. Although this can achieve better implants and placement plans, it does not help with real-time placement depending on the condition of the patient. Therefore, this study presented a method that provides real-time information on the optimal placement method considering the patient’s implant placement location, bone condition, and distance from surrounding teeth.

The stress generated in the cortical bone and implant because of the masticatory force of the implant was analyzed via structural analysis. The length and thickness of the implant, Young’s modulus of the cancellous bone, front–rear angle, left–right angle, and thickness of the cortical bone were used as the influencing design variables, and they were set as 11.5 mm, 4.0 mm, 7.0 GPa, 0 degree, 0 degree, and 1.5 mm, respectively, to ensure the same conditions. The teeth were divided into three categories: anterior, premolar, and posterior teeth.

A frictional contact was applied after the implant and bone were completely attached, and the results were confirmed when the implant was fully installed. Subsequently, the analysis was conducted by applying a force from the top to the bottom based on a masticatory force size of 100 N.

The analysis results indicated that less stress occurred from the top of the cortical bone to the bottom of the cancellous bone, and the highest force occurred at the first thread where the implant and bone met. Among the several implant variables, the stress in the bone became widely distributed with an increase in the length and diameter of the implant, which decreased the stress in the cortical bone. The force acting on the cortical bone was distributed to the cancellous bone with an increase in the Young’s modulus of the cancellous bone, thereby resulting in a decrease in stress. The force received by the implant was not directed vertically downward in the local direction of the implant when changing its installation angle. Thus, stress increased with an increase in the angle, and the greatest force occurred at the first curved part, not at the first position of the implant or screw thread. The cortical bone thickness did not show a clear trend based on the change. The force was distributed and the stress decreased with an increase in the thickness of the cortical bone; however, the overall Young’s modulus and stress increased with a decrease in the size of the cancellous bone. No clear trend was observed because of the two offsetting effects, which lowered accuracy while generating ROM data; an analysis was conducted considering this point. Comparing the difference between the minimum and maximum stresses within the design variables used confirmed that the order of influence was as follows: Young’s modulus of the cancellous bone > implant thickness > front–rear angle > left–right angle > implant length.

These results were used to create a parametric ROM for designing a reduced model that responded to the changes in the parameters. The parametric ROM production was evaluated using the polynomial, SVR, GARS, and DFFN methods. These methods were tested and the the highest accuracy method was selected. COD was used to evaluate the approximation quality of the polynomial regression model, and COP was used to evaluate the prognostic quality.

Compared to the other methods, GARS showed an average COD accuracy of 99% and COP accuracy of 83%. For the cortical bone stress, the parametric ROM using GARS was selected because it showed an accuracy of COD of over 97% and a COD of over 88% under all three tooth conditions. The data-driven parametric ROM obtained using this, i.e., a 1-D CAE solver, was created to calculate the expected stress in the cortical bone by determining the values when inputting the values for each variable.

The calculated values were loaded into OnDemand3D software, and the position and angle of the implant, as determined from the data obtained through the video, were applied to the parameter column. The 1-D CAE solver was automatically calculated when the implant length, diameter, and Young’s modulus of the cancellous bone were entered according to the patient’s condition; subsequently, the results were derived. If the maximum stress generated in the cortical bone was more than 40 MPa, the implant was displayed in red; if it was less than that, the implant was displayed in green. The stress value under the condition was derived such that stability could be confirmed visually.

In contrast to the existing method of establishing an implantation plan based on the doctor’s judgment, the proposed ROM method ensures objectivity. Further, various virtual placement methods can be considered because the stress at the placement location is determined in real time using data obtained from image data and existing data according to the placement location. However, the proposed method has a limitation in that an implantation plan has been established considering only six variables. In future research, a ROM with additional variables will be produced to create a safer and more comprehensive implantation plan. Thus, the parametric ROM-based method for analyzing implant placement plans is expected to be used as an important module in building dental digital twins.

## Figures and Tables

**Figure 1 bioengineering-11-00084-f001:**
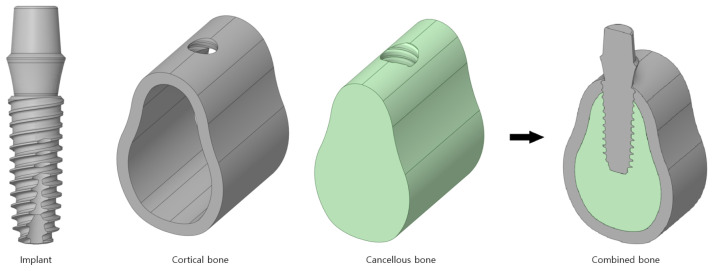
The 3D CAD model of the implant body.

**Figure 2 bioengineering-11-00084-f002:**
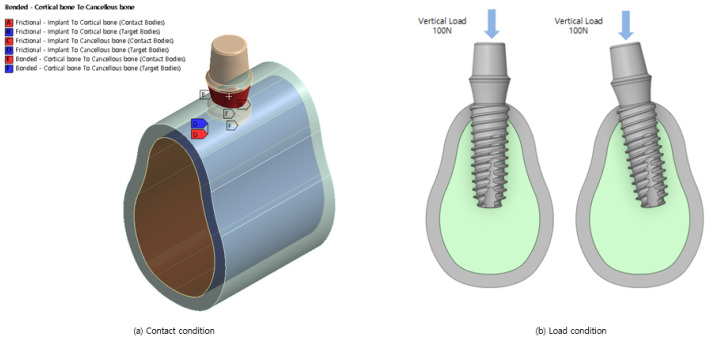
Boundary conditions.

**Figure 3 bioengineering-11-00084-f003:**
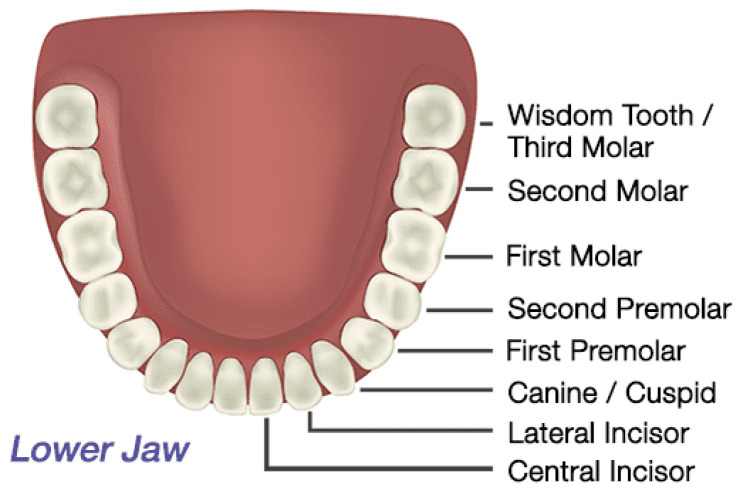
Lower jaw teeth.

**Figure 4 bioengineering-11-00084-f004:**
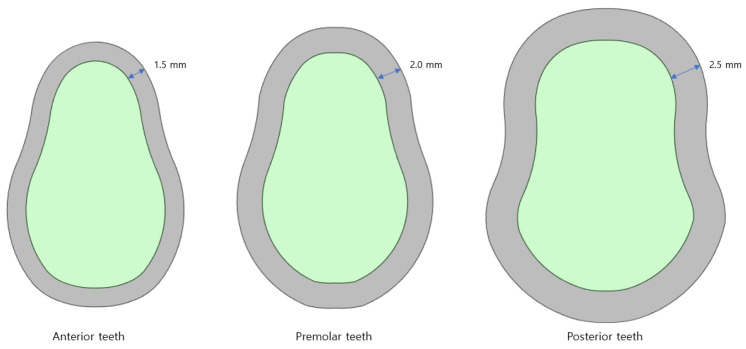
Bones of the anterior, premolar, and posterior teeth.

**Figure 5 bioengineering-11-00084-f005:**
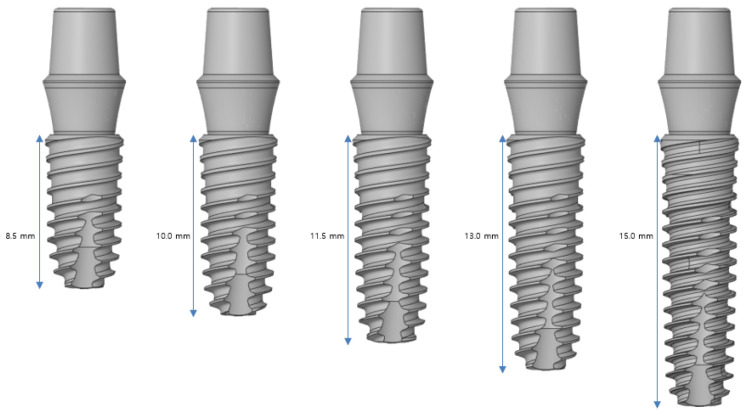
Variable 1—Implant length.

**Figure 6 bioengineering-11-00084-f006:**
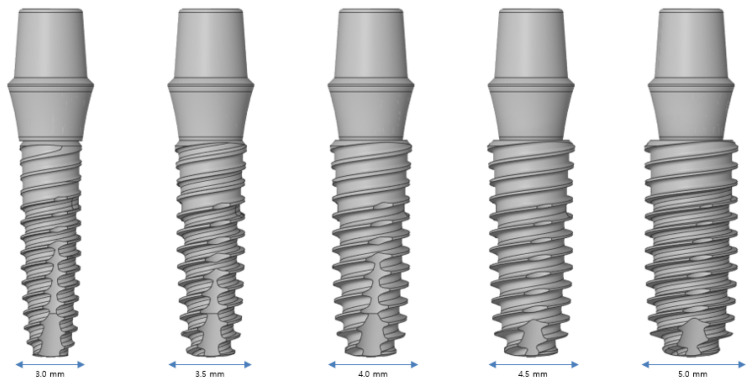
Variable 2—Implant diameter.

**Figure 7 bioengineering-11-00084-f007:**
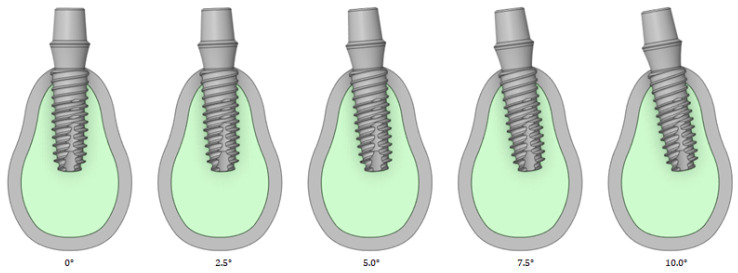
Variable 4—Implant front–rear angle.

**Figure 8 bioengineering-11-00084-f008:**
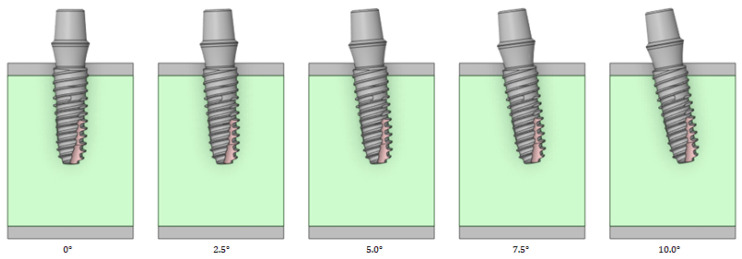
Variable 5—Implant left–right angle.

**Figure 9 bioengineering-11-00084-f009:**
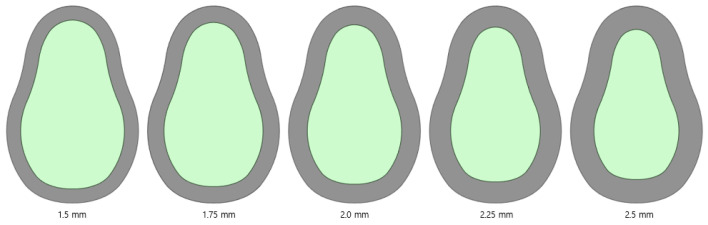
Variable 6—Cortical bone thickness.

**Figure 10 bioengineering-11-00084-f010:**
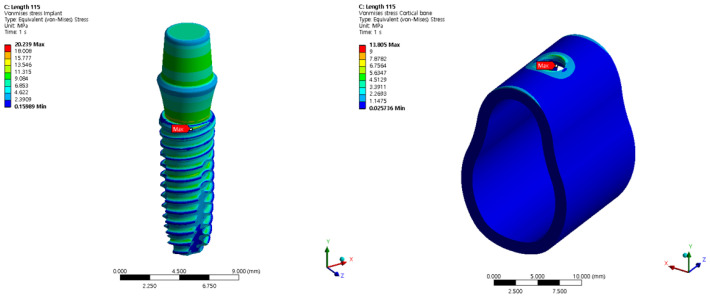
The von Mises stress for the implant and cortical bone.

**Figure 11 bioengineering-11-00084-f011:**
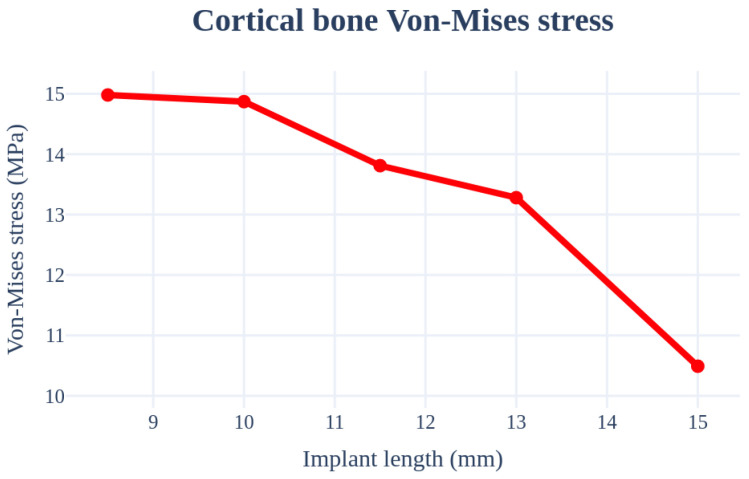
Chart of cortical bone von Mises stress value (Implant length).

**Figure 12 bioengineering-11-00084-f012:**
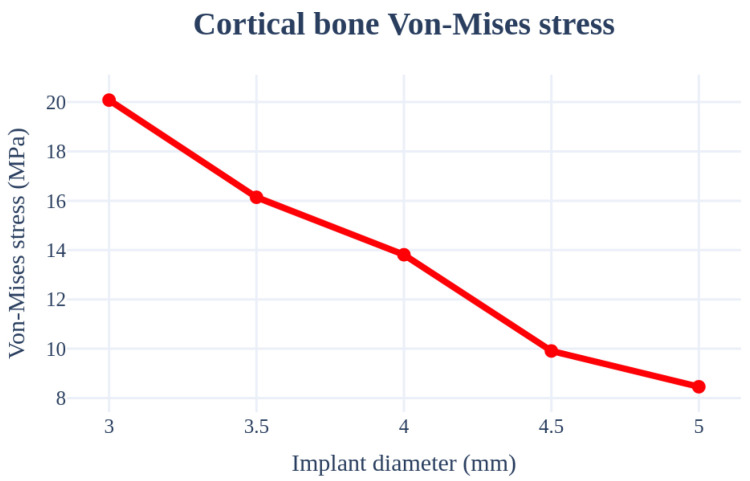
Chart of the cortical bone von Mises stress value (Implant diameter).

**Figure 13 bioengineering-11-00084-f013:**
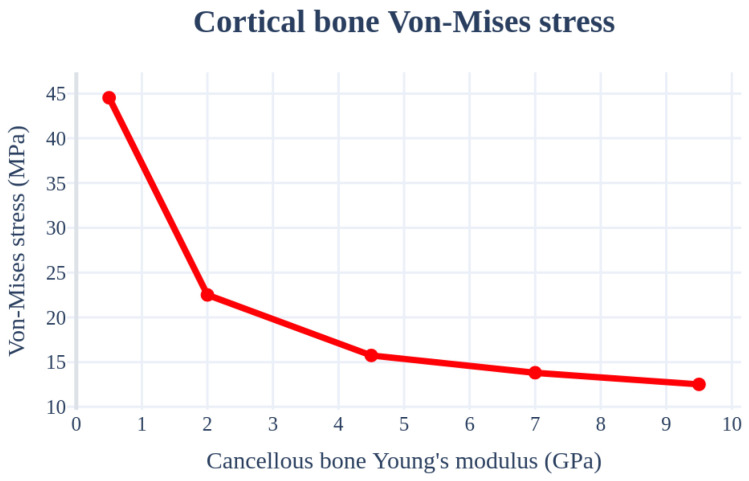
Chart of the cortical bone von Mises stress value (Young’s modulus of the cancellous bone).

**Figure 14 bioengineering-11-00084-f014:**
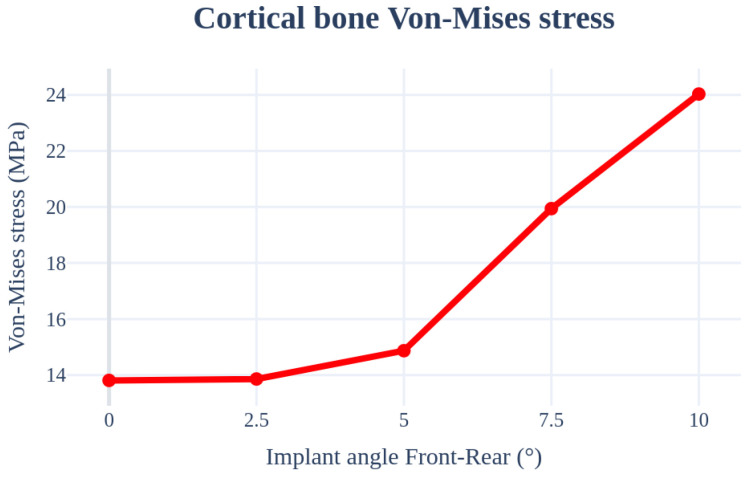
Chart of the cortical bone von Mises stress value (Implant angle front–rear).

**Figure 15 bioengineering-11-00084-f015:**
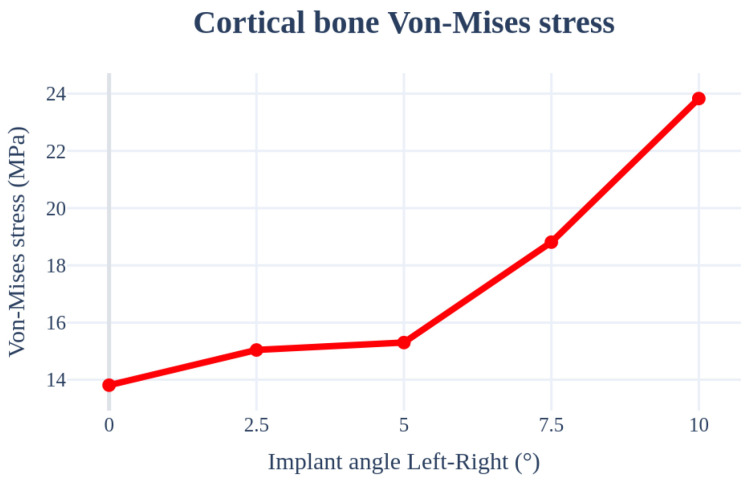
Chart of the cortical bone von Mises stress value (Implant angle left–right).

**Figure 16 bioengineering-11-00084-f016:**
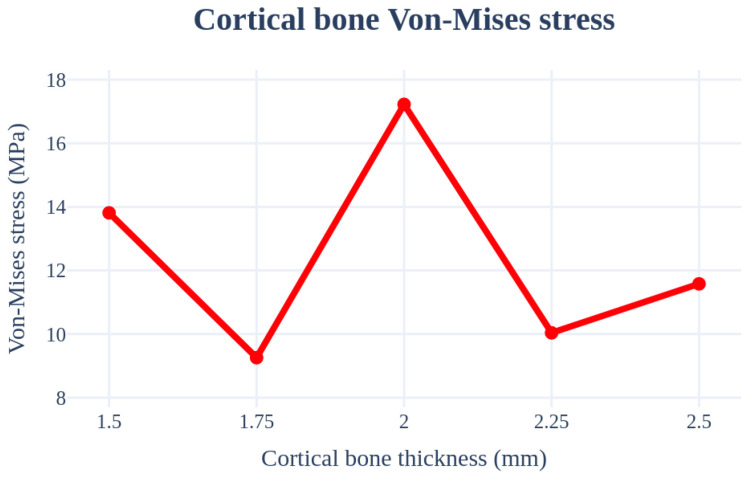
Chart of the cortical bone von Mises stress value (Cortical bone thickness).

**Figure 17 bioengineering-11-00084-f017:**
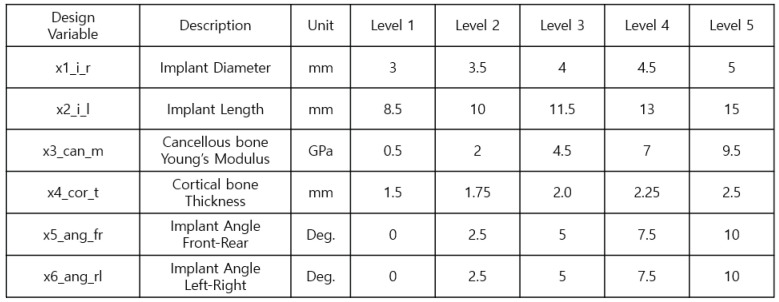
Controllable parameters and their associated levels.

**Figure 18 bioengineering-11-00084-f018:**
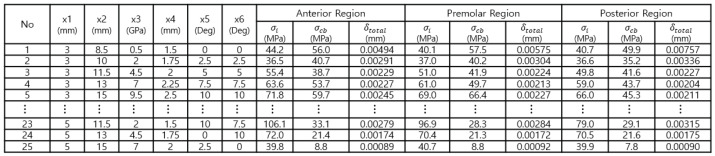
Taguchi orthogonal array level 25.

**Figure 19 bioengineering-11-00084-f019:**
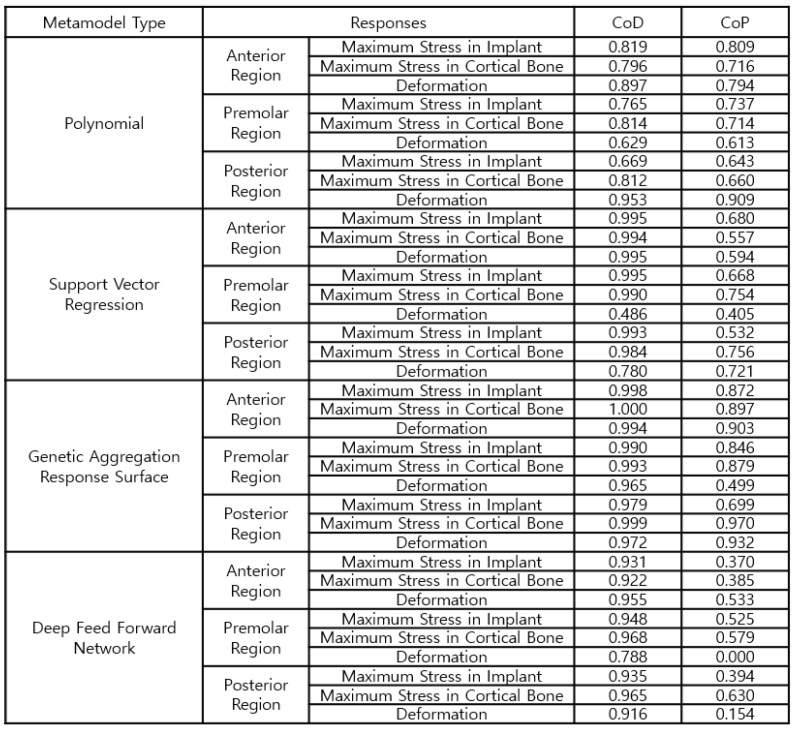
Configurations of the high-fidelity parametric ROM.

**Figure 20 bioengineering-11-00084-f020:**
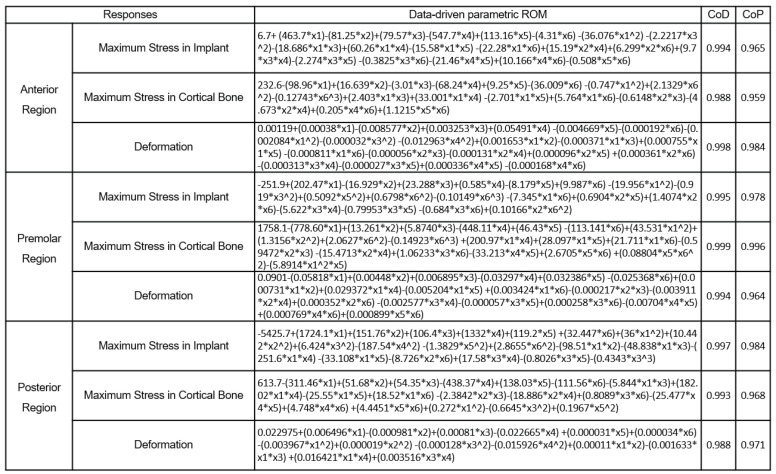
Data-driven parametric ROM to predict the anterior, premolar, and posterior regions at three different locations.

**Figure 21 bioengineering-11-00084-f021:**
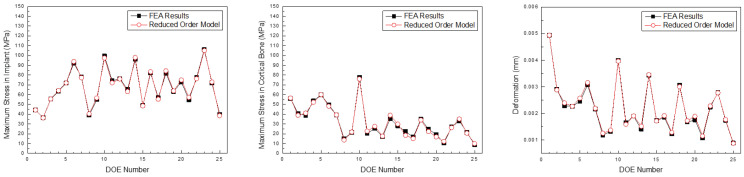
Comparison of the maximum stress in the implant, cortical bone, and deformation predicted by the parametric ROM and FEA results (Anterior teeth).

**Figure 22 bioengineering-11-00084-f022:**
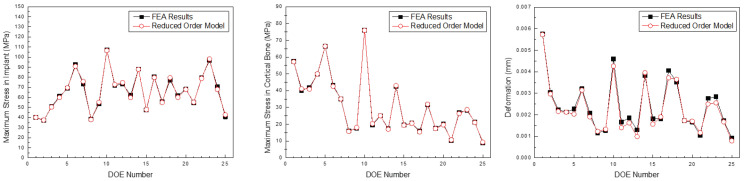
Comparison of the maximum stress in the implant, cortical bone, and deformation predicted by the parametric ROM and FEA results (Premolar teeth).

**Figure 23 bioengineering-11-00084-f023:**
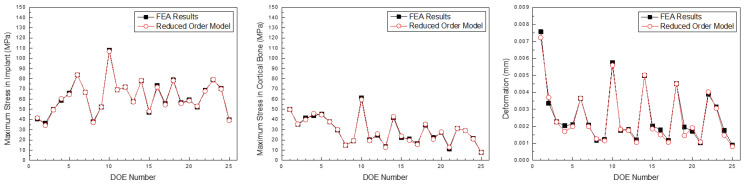
Comparison of the maximum stress in the implant, cortical bone, and deformation predicted by the parametric ROM and FEA results (Posterior teeth).

**Figure 24 bioengineering-11-00084-f024:**
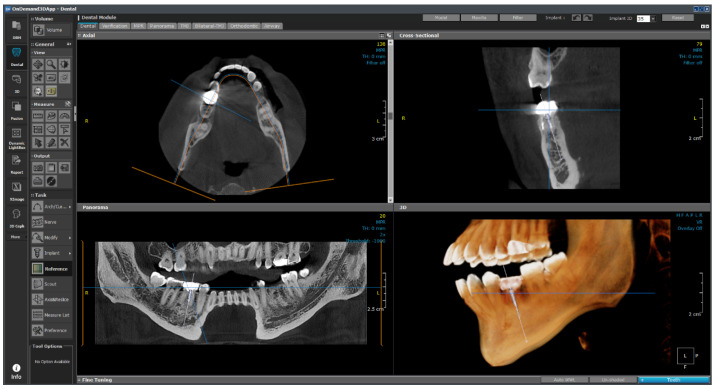
Layout of the dental module.

**Figure 25 bioengineering-11-00084-f025:**
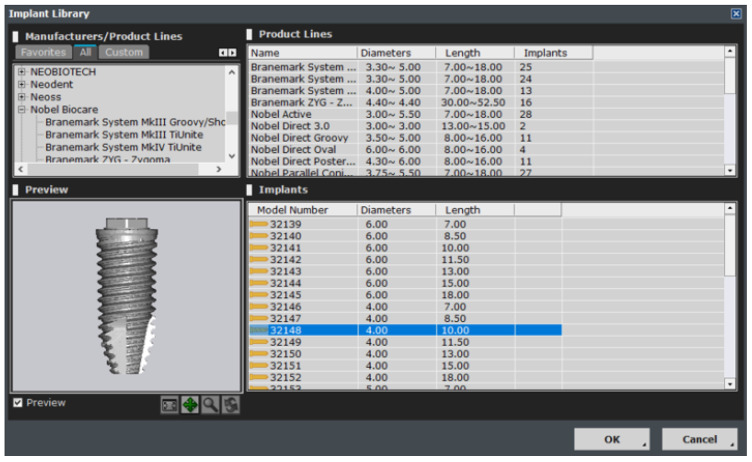
Implant library.

**Figure 26 bioengineering-11-00084-f026:**
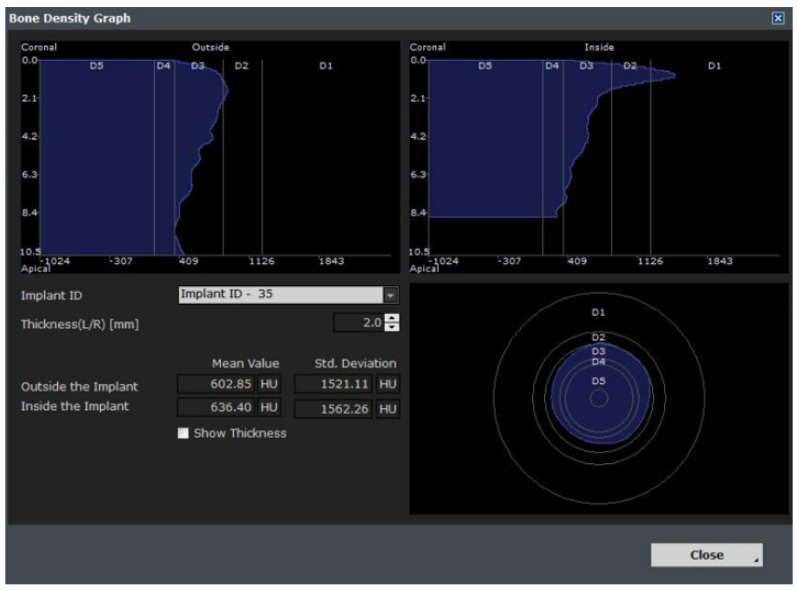
Bone information.

**Figure 27 bioengineering-11-00084-f027:**
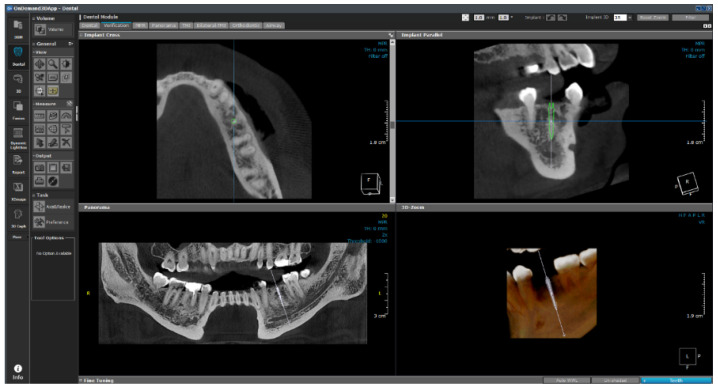
OnDemand3D featuring implant planning and surgical guide design.

**Figure 28 bioengineering-11-00084-f028:**
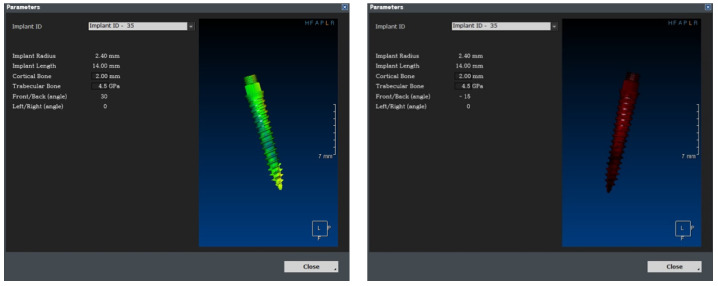
Real-time implant FEA results using 1-D CAE solver.

**Table 1 bioengineering-11-00084-t001:** Material properties.

Materials	Young’s Modulus (GPa)	Poisson’s Ratio
Implant	105	0.37
Cortical bone	13	0.3
Cancellous bone	7	0.3

**Table 2 bioengineering-11-00084-t002:** Variable 3—Young’s modulus of the cancellous bone.

Materials	Young’s Modulus (GPa)	Poisson’s Ratio
Cancellous bone Type-1	0.5	0.3
Cancellous bone Type-2	2.0	0.3
Cancellous bone Type-3	4.5	0.3
Cancellous bone Type-4	7.0	0.3
Cancellous bone Type-5	9.5	0.3

**Table 3 bioengineering-11-00084-t003:** Results of the cortical bone von Mises stress values (Implant length).

von Mises Stress	8.5 mm	10.0 mm	11.5 mm	13.0 mm	15.0 mm
Cortical bone	14.98 MPa	14.87 MPa	13.81 MPa	13.28 MPa	10.49 MPa

**Table 4 bioengineering-11-00084-t004:** Results of the cortical bone von Mises stress values (Implant diameter).

von Mises Stress	3.0 mm	3.5 mm	4.0 mm	4.5 mm	5.0 mm
Cortical bone	20.08 MPa	16.14 Mpa	13.81 Mpa	9.91 Mpa	8.46 Mpa

**Table 5 bioengineering-11-00084-t005:** Results of the cortical bone von Mises stress values (Young’s modulus of the cancellous bone).

von Mises Stress	Type-1	Type-2	Type-3	Type-4	Type-5
Cortical bone	44.53 MPa	22.50 MPa	15.74 MPa	13.81 MPa	12.51 MPa

**Table 6 bioengineering-11-00084-t006:** Results of the cortical bone von Mises stress values (Implant angle front–rear).

von Mises Stress	0°	2.5°	5°	7.5°	10°
Cortical bone	13.81 MPa	13.86 MPa	14.87 MPa	19.94 MPa	24.03 MPa

**Table 7 bioengineering-11-00084-t007:** Results of cortical bone von Mises stress values (Implant angle left–right).

von Mises Stress	0°	2.5°	5°	7.5°	10°
Cortical bone	13.81 MPa	15.04 MPa	15.30 MPa	18.81 MPa	23.83 MPa

**Table 8 bioengineering-11-00084-t008:** Results of cortical bone von Mises stress values (Cortical bone thickness).

von Mises Stress	1.5 mm	1.75 mm	2.0 mm	2.25 mm	2.5 mm
Cortical bone	13.81 MPa	9.26 MPa	17.22 MPa	10.04 MPa	11.58 MPa

**Table 9 bioengineering-11-00084-t009:** Comparison of Solving Time for FEA and Parametric ROM.

Solution	FEA	Parametric ROM
Solving time	2 h 25 min	1 s

## Data Availability

The data used to support the finding of this study are included within the article.

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
