# Peer review of "Toward Digital Twin Development for Implant Placement Planning Using a Parametric Reduced-Order Model"

_bioengineering, 2024, doi:10.3390/bioengineering11010084_

Round 1

Reviewer 1 Report

Comments and Suggestions for Authors

Please include numerical results in the summary section.

In the introduction section, instead of simple references, include which similar studies were conducted by whom and the scientific reasons behind the results obtained.

The results under the headings given in the results and discussion section should be based on scientific reasons and reference should be made to the scientific reasons in the literature. The discussion must be developed.

Comments on the Quality of English Language

Minor editing of English language required

Author Response

Thank you very much for taking the time to review this manuscript. Please find the detailed responses in the attachment. 

Reviewer 2 Report

Comments and Suggestions for Authors

1. The way of reporting is very weak. Authors have to decide what kind of article type they want to present? Is it a review paper or original paper? Please define the type of article and significantly improve the reporting. Authors have to add a typical sections for defined type of article.

2. After determining the type of article, Authors have to also correct the abstract.

3. Authors have to describe why this study is novel and important (rationale the study). Please add a paragraph before the aim of study in Introduction.

4. Please define clear aim of the study in the abstract and at the end of Introduction.

5. Please use the full term before first use of abbreviation within abstract and manuscript body.

6. Authors cited a lot of old literature. Please do not cite articles older than 10 years because mostly they are outdated. This discipline is developing very fast. Therefore Authors have to use the latest literature. Authors should extend the list of references above 30 positions. I suggest to use the latest and reliable literature strongly related to the topic (the citation of mentioned articles is not obligatory):

-Abarno S, Gehrke AF, Dedavid BA, Gehrke SA. Stress distribution around dental implants, generated by six different ceramic materials for unitary restoration: An experimental photoelastic study. Dent Med Probl. 2021;58(4):453–461. doi:10.17219/dmp/135997

-Dommeti VK, Pramanik S, Roy S. Design of customized coated dental implants using finite element analysis. Dent Med Probl. 2023;60(3):385–392. doi:10.17219/dmp/142447

-Tamrakar SK, Mishra SK, Chowdhary R, Rao S. Comparative analysis of stress distribution around CFR-PEEK implants and titanium implants with different prosthetic crowns: A finite element analysis. Dent Med Probl. 2021;58(3):359–367. doi:10.17219/dmp/133234

-Silveira MPM, Campaner LM, Bottino MA, Nishioka RS, Borges ALS, Tribst JPM. Influence of the dental implant number and load direction on stress distribution in a 3-unit implant-supported fixed dental prosthesis. Dent Med Probl. 2021;58(1):69–74. doi:10.17219/dmp/130847

7. Please add a legend of used abbreviations below each table and figure.

Comments on the Quality of English Language

The language of the manuscript has to be revised by native speaker after correction.

Author Response

(The authors gave the same response as above.)

Reviewer 3 Report

Comments and Suggestions for Authors

This manuscript presents a novel method for dental implant planning using parametric reduced order modeling (ROM). The approach shows promise for enabling real-time assessment of implant stability based on variables like bone density and implant angle. However, there are some major issues that need to be addressed before considering publication.

Major Weak Points

  1. The English language quality needs significant improvement throughout, with many grammatical errors, awkward phrasings, and unclear sentences. Professional editing assistance is highly recommended.
  2. The literature review and background fail to sufficiently cite and discuss related prior work on finite element modeling for implant planning and biomechanics. The authors need to better position their contributions in context of previous studies.
  3. Details on the methodology and validation are lacking. More specifics are needed on the FE modeling parameters, assumptions, boundary conditions, and solution methods. Validation cases showing accuracy of the ROM predictions compared to FE analyses for additional scenarios beyond what is presented would significantly strengthen the study.
  4. The clinical/practical utility of the proposed workflow needs better justification. The authors claim it enables "real-time" assessment but the concept of real-time needs clarification in this context, and quantitative metrics demonstrating computational speedup compared to traditional FE analysis should be provided.

Additional Weak Points

  • More details needed on OnDemand3D software implementation and user interface for utilizing the ROM
  • Conclusions do not provide a critical assessment of limitations or future work
  • Some figures are missing axes labels or units
  • References are appropriately cited in text, but reference list is incomplete

Recommendation

In its current form, I would recommend rejecting the paper and encouraging the authors to address the major issues identified, as well as thoroughly edit the paper for English language usage, before resubmitting to Bioengineering. With significant revisions, this work could present a compelling study for publication.

Comments on the Quality of English Language Extensive editing of English language required

Author Response

(The authors gave the same response as above.)

Round 2

Reviewer 2 Report

Comments and Suggestions for Authors

The manuscript has been strongly improved. Authors should consider to cite  the following and latest literature which is very important to this topic:

-Abarno S, Gehrke AF, Dedavid BA, Gehrke SA. Stress distribution around dental implants, generated by six different ceramic materials for unitary restoration: An experimental photoelastic study. Dent Med Probl. 2021;58(4):453–461. doi:10.17219/dmp/135997

-Dommeti VK, Pramanik S, Roy S. Design of customized coated dental implants using finite element analysis. Dent Med Probl. 2023;60(3):385–392. doi:10.17219/dmp/142447

Comments on the Quality of English Language

Minor editing of English language required.

Reviewer 3 Report

Comments and Suggestions for Authors

After reviewing the initial submission and revised version of the manuscript "Towards digital twin development for implant placement planning using parametric reduced-order model", I recommend accepting this paper for publication. The authors have adequately addressed the feedback from the first round and significantly improved the quality and clarity of the article in the updated version.

Comments on the Quality of English Language

Minor mistakes